# Nematophagous fungus *Arthrobotrys oligospora* mimics olfactory cues of sex and food to lure its nematode prey

Yen-Ping Hsueh[1,2], Matthew R Gronquist[3], Erich M Schwarz[4], Ravi David Nath[1], Ching-Han Lee[2], Shalha Gharib[1], Frank C Schroeder[5,6], Paul W Sternberg[1]*

[1]Division of Biology and Biological Engineering, Howard Hughes Medical Institute, California Institute of Technology, Pasadena, United States; [2]Institute of Molecular Biology, Academia Sinica, Taipei, Taiwan; [3]Department of Chemistry, State University of New York at Fredonia, Fredonia, United States; [4]Department of Molecular Biology and Genetics, Cornell University, Ithaca, United States; [5]Boyce Thompson Institute, Cornell University, Ithaca, United States; [6]Department of Chemistry and Chemical Biology, Cornell University, Ithaca, United States

**Abstract** To study the molecular basis for predator-prey coevolution, we investigated how *Caenorhabditis elegans* responds to the predatory fungus *Arthrobotrys oligospora*. *C. elegans* and other nematodes were attracted to volatile compounds produced by *A. oligospora*. Gas-chromatographic mass-spectral analyses of *A. oligospora*-derived volatile metabolites identified several odors mimicking food cues attractive to nematodes. One compound, methyl 3-methyl-2-butenoate (MMB) additionally triggered strong sex- and stage-specific attraction in several *Caenorhabditis* species. Furthermore, when MMB is present, it interferes with nematode mating, suggesting that MMB might mimic sex pheromone in *Caenorhabditis* species. Forward genetic screening suggests that multiple receptors are involved in sensing MMB. Response to fungal odors involves the olfactory neuron AWCs. Single-cell RNA-seq revealed the GPCRs expressed in AWC. We propose that *A. oligospora* likely evolved the means to use olfactory mimicry to attract its nematode prey through the olfactory neurons in *C. elegans* and related species.

*For correspondence: pws@ caltech.edu

## Introduction

Predation, in which individuals of one species (predators) kill and consume the biomass of individuals of another species (prey) (*Abrams, 2000*), imposes a strong selective pressure on both predators and prey. To minimize the risk of being eaten, prey have often evolved specific behaviors and strategies such as camouflage, avoidance, mimicry, and tonic immobility to increase their chance of survival. In turn, predators also evolved enhanced predatory strategies to secure food sufficient to survive and reproduce, giving rise to an evolutionary arms race between predator and prey (*Dawkins and Krebs, 1979*).

While most carnivores are animals, examples of predacious plants and fungi exist. Venus flytraps, sundews and pitcher plants acquire some or most of their nutrients from trapping and consuming insects, a feature that evolved in response to nitrogen limitation (*Darwin, 1875*). Many fungi that grow in nitrogen-poor environments have likewise evolved carnivorism and nematodes, being the most numerically abundant animals on earth, conveniently became the prey (*Barron, 1977*; *Nordbring-Hertz, 1988*). This predatory lifestyle has independently evolved multiple times among different fungal lineages including Zygomycetes, Ascomycetes and Basidiomycetes (*Barron, 1977*; *Liou and Tzean, 1997*; *Yang et al., 2007*). Nematode-trapping fungi depend on their elaborate traps to prey

on nematodes. However, the majority of the nematode-trapping fungi do not constitutively generate these trapping structures; trap-morphogenesis is only triggered by the presence of nematodes (*Pramer and Stoll, 1959*). This suggests that trap-formation might be a highly energy-consuming process and that, to conserve energy, these fungi have evolved to sense signals from nematodes, which indicate the presence of prey. One such signal is the group of nematode pheromones, ascarosides. These pheromones regulate various aspects of behavior and development in *C. elegans* and are sufficient to induce trap-morphogenesis in several species of nematode-trapping fungi (*Hsueh et al., 2013*). Since the nematode-trapping fungi have clearly evolved the ability to eavesdrop on nematode communication, we wondered whether the nematodes also sense and respond to their fungal predators.

*C. elegans* is known to respond to pathogenic bacteria. For example, *Pseudomonas aeruginosa* triggers an aversive learning behavior, while another pathogen *Bacillus nematocida* attracts *C. elegans* (*Niu et al., 2010*; *Zhang et al., 2005*). Bacterial secondary metabolites are able to modulate the signaling and the protective lawn-avoidance behavior in *C. elegans*, demonstrating that the sensory neurons of *C. elegans* are critical for its defensive responses against pathogens (*Meisel et al., 2014*; *Pradel et al., 2007*; *Zhang et al., 2005*). By contrast, little is known about how *C. elegans* responds to its natural predators such as the nematode-trapping fungi, with only an early study showing that the nematode *Panagrellus redivivus* was attracted to a species of nematode-trapping fungi (*Balan and Gerber, 1972*). Many prey are known to begin fleeing when they sense a possible predator, while many predators are also known to attract their prey (*Haynes et al., 2002*; *Rosier, 2011*). Therefore, we investigated the behavioral and molecular basis for how *C. elegans* responds to *Arthrobotrys oligospora*, one of the most common nematode-trapping fungal species and one that we had previously found to eavesdrop on ascarosides produced by *C. elegans* (*Hsueh et al., 2013*).

Here, we show that *C. elegans* and other nematodes are attracted to *A. oligospora* and the attraction observed for *C. elegans* is mediated by the two AWC olfactory neurons. We identified several odorants produced by *A. oligospora* and found that many of the odorants attracted *C. elegans*, and appear to represent olfactory mimics of the food and sex cues. Through genetic screens and single-cell transcriptome analyses of the AWC neuron, we identified the potential chemosensory receptors that were expressed in this neuron, which is involved in sensing some of these fungal odors. Our study shows that in order to catch its nematode prey, *A. oligospora* has evolved to lure the nematodes by producing olfactory mimicry of the food and sex cues that are attractive to *Caenorhabditis* nematodes.

## Results

### Nematode-trapping fungus *A. oligospora* attracts *C. elegans* and other nematodes

To study how *C. elegans* responds to one of its natural predators, the nematode-trapping fungus *A. oligospora*, we tested the chemotaxis behavior of *C. elegans* in the presence of the fungal culture. We designed a four-point chemotaxis assay. In this assay, *A. oligospora* cultures were grown in two quadrants of a plate, following which synchronized adult *C. elegans* were placed in the middle of the plates and allowed to crawl freely until they reached a source of sodium azide in each quadrant that paralyzed the worms (*Figure 1A*). The number of immobilized worms in each of the quadrants was counted and their chemotaxis index (*Bargmann et al., 1993*) was determined as described in *Figure 1A*. We found that the adult *C. elegans* were consistently attracted to *A. oligospora*, suggesting that the fungus might lure the worms by producing compounds that are attractive to them (*Figure 1B*). This strong attraction was only observed in the adult animals; dauers and the L1 larvae exhibit much weaker attraction to *A. oligospora* (*Figure 1B*). *C. elegans* moves at a similar speed on *A. oligospora* culture; thus we think the attraction is unlikely due to a sedative secreted by the fungus or a mechanical cue that retained the worms in the fungal quadrants (*Figure 1C*).

To investigate whether the attraction towards *A. oligospora* is a conserved behavior observed across different nematode species or if it is specific to *C. elegans*, we tested the chemotaxis behavior of eight additional *Caenorhabditis* species (*Félix et al., 2014*). We observed significant attraction toward *A. oligospora* in six other *Caenorhabditis* species, though not in *C. angaria* and *C. castelli* (*Figure 1D*). We thus further tested three additional nematode species that are more

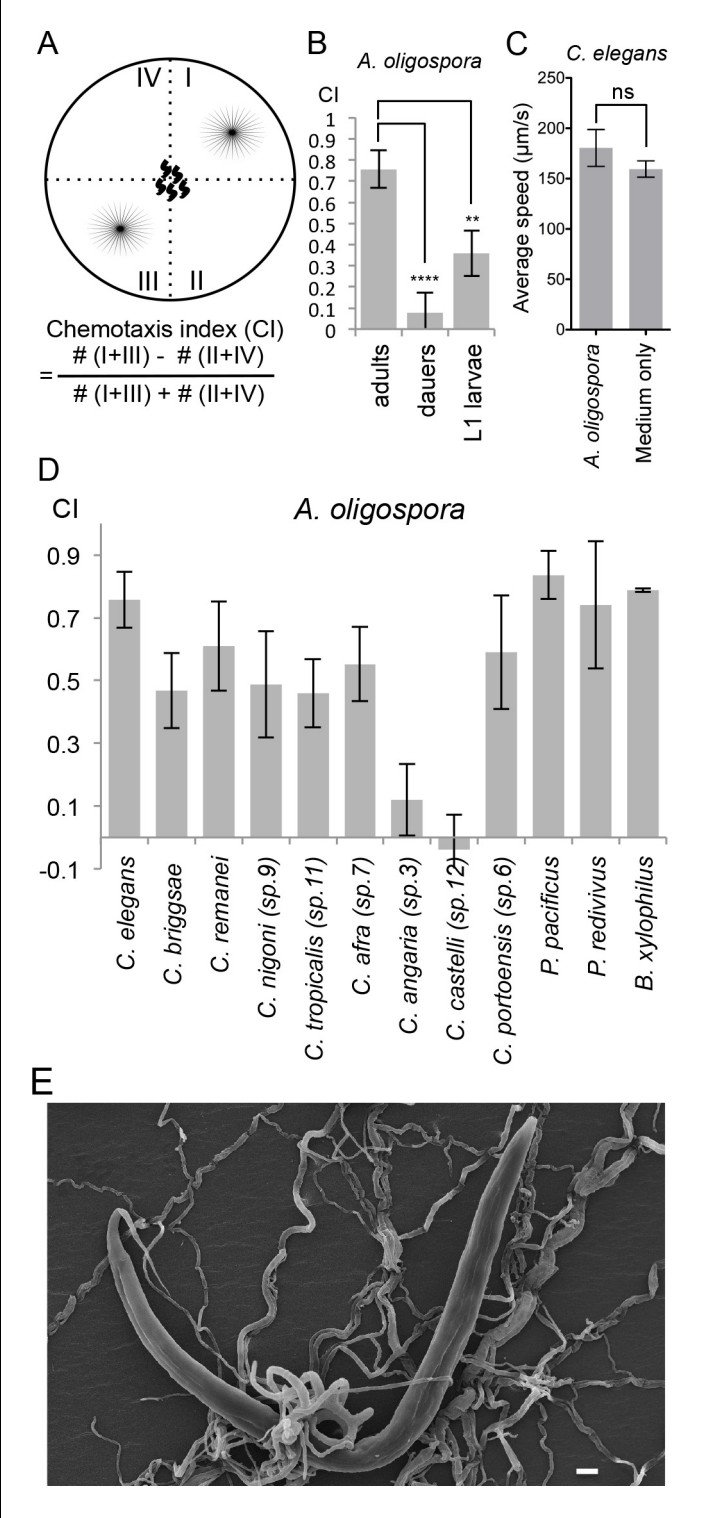

**Figure 1.** *A. oligospora* attracts many nematode species, including *C. elegans*. (**A**) Schematic representation of the 4-point chemotaxis assays and the formulation used to calculate the chemotaxis index (CI). (**B**) *C. elegans* adults are attracted to *A. oligospora*, but the dauers and the L1 larvae have much weaker attraction (Mean ± SD, n = 4–12 trials). (**C**) Speed of wild-type *C. elegans* on *A. oligospora* culture or medium only control. Locomotion of the animals was tracked by the commercial WormLab system (MBF bioscience). (**D**) Many *Caenorhabditis* nematodes and the more distant *Panagrellus redivivus, Pristionchus pacificus,* and *Bursaphelenchus xylophilus*
*Figure 1 continued on next page*

*Figure 1 continued*

were attracted to *A. oligospora* (Mean ± SD, n = 4–12 trials). (**E**) SEM image of *C. elegans* trapped in *A. oligospora*. Scale bar = 10 μm.

phylogenetically distant from *C. elegans*: *Pristionchus pacificus*, *Panagrellus redivivus*, and *Bursaphelenchus xylophilus*. Like *C. elegans*, *P. pacificus* and *P. redivivus* are both free-living bacterivorous nematodes, whereas *B. xylophilus* is a fungal feeder and a parasite of pine trees (*Futai, 2013*; *Sommer, 2006*; *Srinivasan et al., 2013*). We found that all three of these additional species were also attracted to *A. oligospora*, indicating that *A. oligospora* produces diffusible compounds that are attractive to a variety of nematodes to help lure the prey and induce trap-morphogenesis (*Figure 1E*).

## *A. oligospora* produces volatile compounds to attract *C. elegans*

To determine whether these diffusible nematode-attracting compounds were soluble or volatile, we designed another version of the chemotaxis assay using three-division Petri dishes. In this assay, *A. oligospora* was grown in one of the three sectors and *C. elegans* was placed in a separate sector, where the worms could only sense the volatile compounds, but not the soluble ones produced by *A. oligospora* (*Figure 2A*). We let the nematodes move freely for 2 hr, after which their chemotaxis index was determined by counting the number of worms that had migrated to the sectors with or without the fungal culture (*Figure 2A*). We still observed significant attraction of *C. elegans* toward *A. oligospora* (*Figure 2B*), suggesting that *A. oligospora* likely produces volatile compound(s) that are attractive to *C. elegans*.

## The olfactory AWCs neurons mediate *A. oligospora*-attraction in *C. elegans*

We next sought to identify the *C. elegans* sensory neurons responsible for sensing odors emanating from *A. oligospora* and for mediating the attraction. There are 32 sensory neurons present in the amphid, phasmid and inner labial organs of *C. elegans*. For some of these neurons, their functions have been studied in detail (*Bargmann, 2006*). For example, the amphid ASE neurons are required for chemotaxis to water-soluble attractants, whereas AWA and AWC neurons mediate chemotaxis to volatile attractants (*Bargmann et al., 1993*; *Bargmann and Horvitz, 1991*).

To determine whether genes known to play a role in chemotaxis are required for sensing *A. oligospora,* we first tested several known mutants for *A. oligospora* attraction. We found that *che-1*, a gene required for ASE neuron identity and function, was dispensable for *A. oligospora* attraction but that genes known to play a role in AWC function, such as *odr-1*, *odr-3*, *tax-2* and *tax-4*, were required (*Figure 2C*). We thus laser-ablated both AWC neurons in *C. elegans* and analyzed these animals individually for chemotaxis to *A. oligospora*. Approximately 70% of the trials of mock-ablated control animals were attracted to *A. oligospora,* whereas only 20% of the trials of the AWC-ablated animals were attracted, indicating that AWCs play a major role in *A. oligospora* attraction (*Figure 2D*).

## Identification of *A. oligospora*-derived odorants that attract *C. elegans*

To understand how *A. oligospora* attracts nematodes, we decided to identify *A. oligospora*-derived odorants. Solid-phase microextraction (SPME) was used to sample headspace volatiles from vials containing medium inoculated with *A. oligospora* and from control vials containing only medium. The samples then were subjected to comprehensive two-dimensional gas chromatographic mass spectral analyses (GC x GC-TOFMS). Five compounds present only in samples containing *A. oligospora* culture were identified: dimethyl disulfide (DMDS), (±)−2-methyl-1-butanol (MB), 2,4-dithiapentane (DTP), methyl 3-methyl-2-butenoate (MMB), and S-methyl thioacetate (SMT; *Figure 3A* and *Supplementary file 1*). To examine whether these *A. oligospora*-derived odorants attracted *C. elegans*, we tested the individual compounds in chemotaxis assays. We found that DMDS was only moderately attractive to *C. elegans*, but that all the other compounds, especially MMB, were highly attractive to *C. elegans* (*Figure 3B*). DMDS has been identified from many bacteria (*Tomita et al., 1987*) and we have also detected DMDS in another two ascomycete fungi, *Aspergillus terreus* and

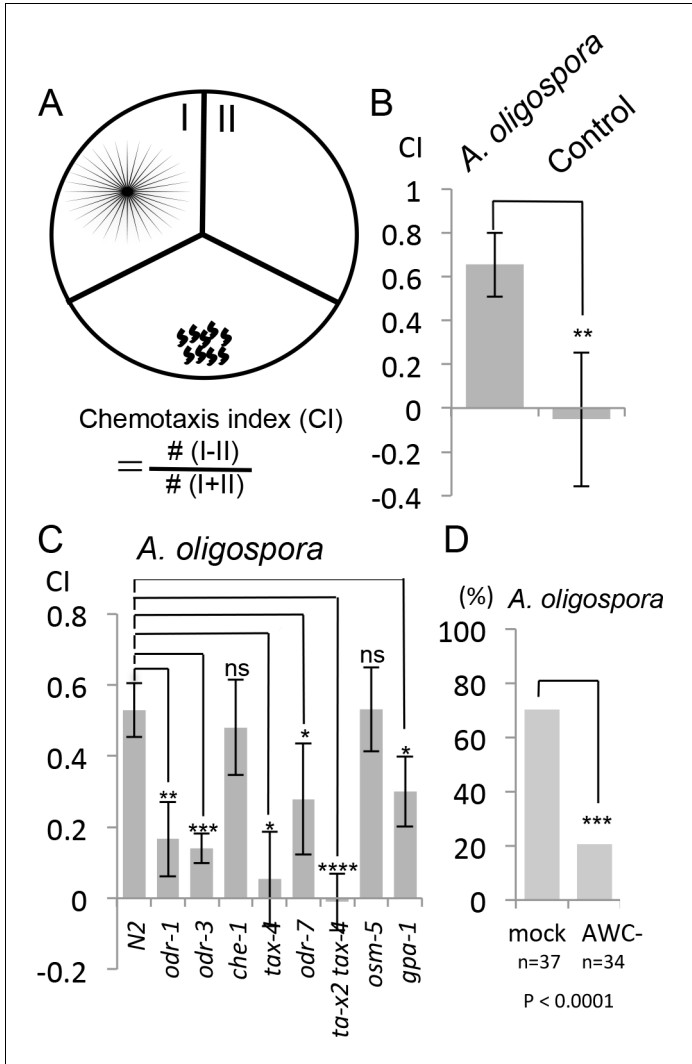

**Figure 2.** *C. elegans* are attracted to volatile compounds produced by *A. oligospora*. (**A**) Schematic representation of the volatile chemotaxis assays in which *A. oligospora* were grown in compartmentalized Petri dishes that contained three sections. The CI was calculated based on the formula presented. (**B**) *C. elegans* adults preferentially migrate to the section that was inoculated with *A. oligospora* (Mean ± SD, n = 6–7 trials). (**C**) *A. oligospora* attraction is reduced in several mutants defective in AWC neuron function (Mean ± SD, n = 3–4 trials). (**D**) Laser ablation of the AWC neurons reduced attraction to *A. oligospora*.

*A. niger* (**Supplementary file 1**). These observations suggest that DMDS is a relatively common odor, and it may be sensed as a food cue by the bacterivorous *C. elegans*. DMDS, MB, and DTP have been previously identified as odors that contribute to truffle aroma (**Díaz et al., 2002**; **Pacioni et al., 2014**; **Pennazza et al., 2013**); we therefore tested *C. elegans* chemotaxis in response to commercial truffle oil. We found that truffle oil was very attractive to *C. elegans* (**Figure 3B**). These results indicate that several *A. oligospora*-derived odorants might mimic food signals to attract nematodes.

## Attraction to *A. oligospora*-derived chemicals diminished upon adaptation to the fungal culture

One of the identified odorants, MMB, was highly attractive to *C. elegans*. When varying doses were tested for elicited responses, MMB attracted *C. elegans* across a $10^6$ fold concentration range (**Figure 3C**). Moreover, when *C. elegans* were given a choice between isoamyl alcohol (IAA, a previously described attractant) and MMB, the worms exhibited a strong preference for MMB (**Figure 3D**).

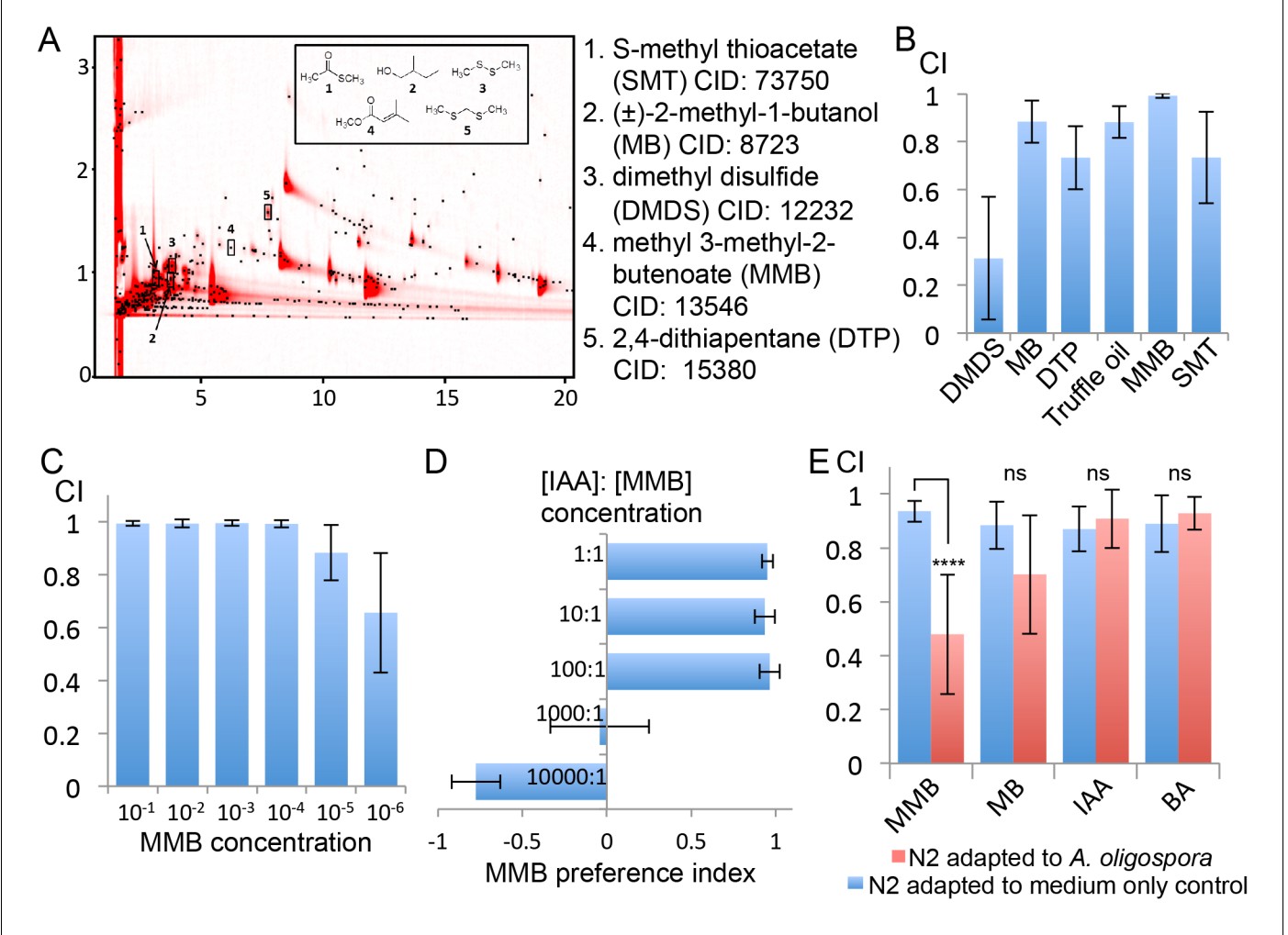

**Figure 3.** GC x GC-TOFMS analyses identified *A. oligospora*-derived odorants that are attractive to *C. elegans*. (A) Representative two-dimensional gas chromatogram obtained from SPME sampling of headspace above *A. oligospora* cultures. The total detector response is indicated by the intensity of red in the chromatogram. Black dots represent discrete elution peaks identified during data processing (shown for a signal-to-noise threshold of 100). The elution peaks labeled 1–5 represent compounds that were detected only in *A. oligospora* cultures, and not in the medium-only controls. (B) Chemotaxis assays of *A. oligospora*-derived odorants. Chemicals were tested at a $10^{-2}$ dilution and truffle oil was tested undiluted (Mean ± SD, n = 6 trials). (C) N2 adults were strongly attracted to MMB across a wide-range of concentrations (Mean ± SD, n = 6 trials). (D) *C. elegans* prefers MMB over isoamyl alcohol (IAA; Mean ± SD, n = 3–7 trials). The MMB preference index was calculated according to the formula in *Figure 1*. (E) Chemotaxis plot showing *C. elegans* chemotactic response to different odorants after 2 hr adaptation to *A. oligospora* culture (red) or the medium-only control (blue; Mean ± SD, n = 4–10 trials).

To examine whether MMB is indeed a genuine molecule naturally produced by *A. oligospora,* and not an artifact of GC-MS analysis, we exposed *C. elegans* to *A. oligospora* cultures for two hours, and measured the chemotaxis behavior of these *A. oligospora*-adapted animals. If the *A. oligospora* cultures were emitting MMB, we expected to subsequently observe significantly decreased attraction toward MMB as a result of olfactory adaptation, a behavior that has been well-characterized in *C. elegans* (*Colbert and Bargmann, 1995*). Indeed, we found that when *C. elegans* were exposed to *A. oligospora* volatiles for two hours, their chemotactic response to MMB was strongly reduced (*Figure 3E*). This reduction in chemotaxis was odorant-specific, as opposed to a global reduction in chemotaxis, as *A. oligospora*-adapted animals were still attracted to IAA and benzealdehyde (BA), two odorants that were not detected in the *A. oligospora* volatiles and are known to attract *C. elegans* (*Bargmann et al., 1993*) (*Figure 3E*).

## *A. oligospora*-derived odorants activate the AWC$^{on}$ neuron

Having demonstrated that the AWC neurons are required for attraction towards *A. oligospora*, we examined whether they could be directly activated by *A. oligospora*-derived odorants. The gene *str-2* encodes a predicted G-protein coupled receptor (GPCR) that is randomly and asymmetrically expressed in one of two bilateral AWC chemosensory neurons, called AWC$^{on}$ (*Troemel et al., 1999*). We thus transgenically expressed the calcium indicator GCaMP6 in AWC by driving it with the *str-2* promoter. We then measured the AWC$^{on}$ activity in response to different odorants via a microfluidic device (the olfactory chip) (*Chronis et al., 2007*). The AWC$^{on}$ neurons are known to respond to odor removal (*Chalasani et al., 2007*), so we tested whether the AWC$^{on}$ neurons responded to the removal of the *A. oligospora*-derived odorants. We observed a strong activity in the AWC$^{on}$ neuron triggered by several *A. oligospora*-derived odorants (*Figure 4A*). To test if the activation of the AWC$^{on}$ neuron was a primary response, we also imaged the AWC$^{on}$ activity in an *unc-13* background. Mutations of *unc-13* impair the priming of presynaptic vesicles for release, either impairing or completely blocking synaptic transmission (*Richmond et al., 1999*). AWC$^{on}$ neurons were still strongly activated by MMB removal in an *unc-13* mutant background, suggesting that this activation is likely to be a direct chemosensory response to MMB (*Figure 4B*). Compared to wild-type animals, the AWC$^-$ mutant (*ceh-36*) had a strong defect in MMB attraction (*Figure 4C*).

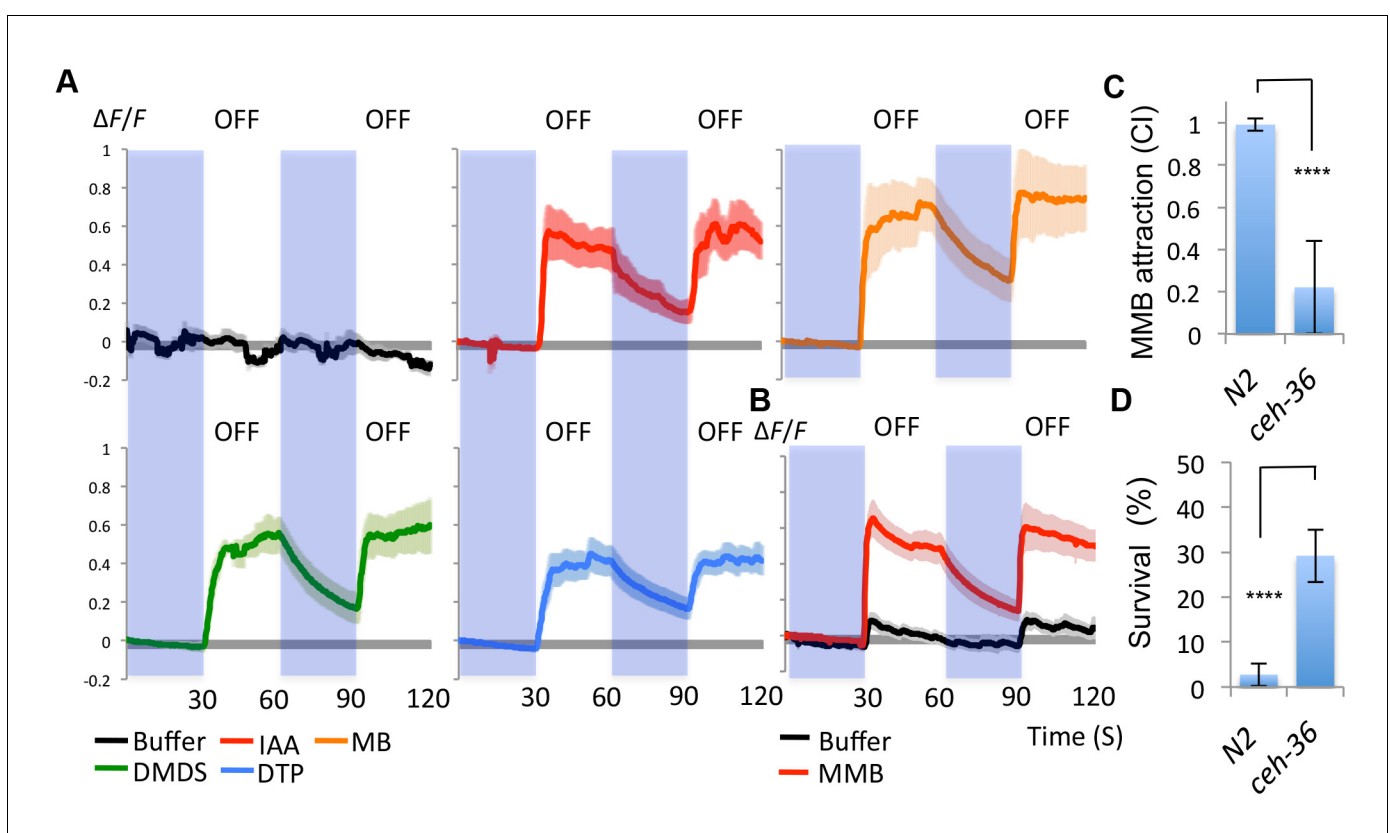

**Figure 4.** AWC$^{on}$ neurons respond to *A. oligospora*-derived odorants. (**A**) GCaMP6 measurements of the AWC$^{on}$ neurons in response to *A. oligospora*-derived odorants ($10^{-3}$) and buffer control (n = 4–7 trials for each odorant tested). (**B**) GCaMP6 measurements of the AWC$^{on}$ neurons in response to MMB in the *unc-13* mutant background (n = 11 and 14 for MMB and buffer control respectively). (**C**) The AWC- mutant is defective in MMB attraction. Chemotaxis response to $10^{-4}$ MMB was measured for the wild-type (N2) or AWC- mutant (*ceh-36*). (**D**) The AWC- mutant is more likely to survive predation by *A. oligospora*. Y axis indicates % of nematodes that were still alive after 16 hr incubation with *A. oligospora* culture (Mean ± SD, n = 10–12 trials).

## AWC-defective mutants exhibit lower predation when encountering *A. oligospora*

Our data suggest that the AWC neurons sense *A. oligospora*-derived odorants and mediate attraction. We thus hypothesized that mutants defective in AWC neuron functioning would exhibit lower predation by *A. oligospora*. To address this hypothesis, *A. oligospora* was inoculated at one side of a 9 cm Petri dish, and *unc-119* nematodes, which are strongly defective in locomotion, were added onto the *A. oligospora* culture to induce trap formation. After traps had emerged from *A. oligospora* hyphae, AWC+ (wild-type) or AWC- (*ceh-36*) animals were then placed at the opposite side of the 9 cm Petri dish and allowed to freely explore. Survival rates of *C. elegans* were determined after an overnight incubation. We found that the majority of the wild-type animals were trapped by *A. oligospora* (with less than 3% of these animals surviving), but ~30% of the AWC- nematodes survived (*Figure 4D*). These results indicate that *C. elegans* with decreased attraction to *A. oligospora* have a better chance to survive under these conditions.

## MMB triggers sex- and developmental stage-specific attraction and interferes with mating in *Caenorhabditis* species

MMB potently attracted *C. elegans* adult hermaphrodites. To investigate why adult hermaphrodites were so attracted to this odor, we further examined the chemotaxis behavior in different developmental stages of *C. elegans*. We found that attraction to MMB was both sex- and developmental stage-dependent, as L1 larvae, dauers, and the male animals did not exhibit strong attraction to this odor (*Figure 5A*). Furthermore, when we tested MMB attraction in other *Caenorhabditis* species, we found that MMB was highly attractive to adult females but repulsive to males in the gonochoristic species such as *C. remanei*, *C. nigoni* (sp. 9), and *C. afra* (sp.7; *Figure 5B*). These observations prompted us to hypothesize that MMB-triggered attraction might be related to sex pheromone-triggered attraction. Therefore, we monitored the mating behavior of *C. afra* in the presence or absence of MMB to examine whether MMB might interfere with mate seeking. We chose this species as we found that it has a highly robust and reproducible mating behavior (*Figure 5C*). When one female and two males were placed on a 3 cm NG plate without food, the nematodes mated within average of ~5 min. However, if 1 μl MMB was placed on the lid of the Petri dishes, in a majority of the trials, the animals failed to mate within the 30 min tracking time, suggesting that MMB interferes with mating (*Figure 5C*). Furthermore, if the female animals were pre-exposed to MMB odor for an hour before testing, mating efficiency also dramatically decreased (*Figure 5C*). In contrast, if mineral

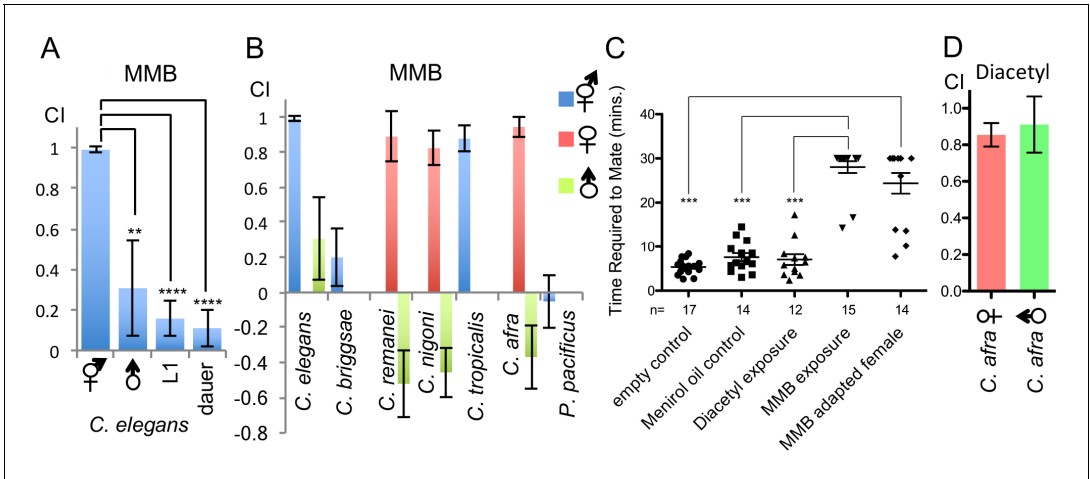

**Figure 5.** MMB triggers a sex- and lifestage-specific attraction in *Caenorhabditis* species and interferes with mating in *C. afra*. (**A**) Responses of *C. elegans* hermaphrodites, males, L1 larvae and dauers to a $10^{-4}$ dilution of MMB (Mean ± SD, n = 4–10 trials). (**B**) Responses of *Caenorhabditis sp.* and *P. pacificus* to a $10^{-4}$ dilution of MMB (Mean ± SD, n = 4–10 trials). (**C**) Mating behavior of *C. afra* was recorded. Y axis indicates the time required for the nematodes to mate; if no mating was observed within the 30 min tracking period, data points were represented as 30 min. (**D**) Chemotaxis of *C. afra* male and female nematodes in response to $10^{-2}$ diacetyl.

oil (solvent control) or diacetyl (another food odor attractive to *C. elegans* and both sexes of *C. afra*; *Figure 5D*) was present, mating was not affected. These results suggested that MMB interfered with mating in the gonochoristic species *C. afra* and are consistent with the hypothesis that MMB might mimic a sex pheromone that specifically and strongly attracted hermaphroditic and female nematodes.

## Genetic screening suggests redundancy for the MMB receptor

We conducted a genetic screen to identify genes that encode a receptor for MMB. The *C. elegans* genome contains ~1300 genes encoding putative chemoreceptors, but only a few of their chemical ligands have been identified (*Thomas and Robertson, 2008*). The diacetyl receptor, *odr-10*, was the first *C. elegans* chemoreceptor identified through a genetic screen and more recently, three pairs of GPCRs (*srbc-64/srbc-66*, *srg-36/srg-37*, and *daf-37/daf-38*) have been shown to encode receptors that sense ascarosides (*Kim et al., 2009*; *Park et al., 2012*; *Sengupta et al., 1996*). The design of our genetic screen followed an approach similar to that used in the screen that identified the diacetyl receptor, *odr-10* (*Sengupta et al., 1996*). In brief, mutagenized animals were given a choice between MMB and IAA (another odor sensed by the AWC neurons). Under this condition, more than 95% of the wild-type N2 animals preferred MMB. We expected that mutants with specific defects in sensing MMB, but with otherwise intact AWC function, would exhibit an abnormal preference for IAA in our screen. After two rounds of behavioral screen enrichment, mutant lines were isolated for further analyses. In total, we isolated thirteen mutants with varying degrees of chemotactic deficiency to MMB. Two mutants (*sy837* and *sy838*) showed dramatic decreases in MMB attraction, whereas eleven other mutant lines only exhibited minor decreases. Diminished MMB attraction seemed to be specific, as attraction to other odorants such as IAA and BA remained at wild-type levels (*Figure 6A*). SNP mapping with the Hawaiian strain revealed the mutations mapped to chromosome X for both strongly defective mutants (*Figure 6B*). Complementation assays revealed that the two mutants failed to complement each other, suggesting that the phenotype was likely caused by a mutation in the same gene (*Figure 6C*). By whole-genome sequencing of the two mutant lines (PS7117 and PS7118), we found that both lines had mutations in the *odr-7* gene, causing premature stop codons at codon 83 and codon 188 respectively (*Figure 6—figure supplement 1*). A genomic rescue line generated with the fosmid WRM0639bF05 that contains the *odr-7* gene restored the chemotaxis defects (*Figure 6—figure supplement 2*). ODR-7 is a nuclear receptor that modulates gene expression in the AWA and AWC neurons, including that of two GPCR genes, *odr-10* and *str-2* (*Colosimo et al., 2003*; *Sengupta et al., 1994*). We identified two independent *odr-7* mutants in our genetic screen but not any other mutants that exhibited a strong reduction in MMB chemotaxis, as one might expect if MMB attraction was mediated via a single chemosensory receptor. Thus, the results from our genetic screen suggest that MMB is more likely to be sensed by multiple redundant receptors in *C. elegans*.

## Single-cell RNA-seq identifies AWC-expressed GPCR candidates that might sense the *A. oligospora*-derived odors

To identify candidate GPCRs expressed in the AWC neurons, we microdissected GFP-labeled AWC$^{on}$ neurons (*str-2*::GFP) and performed single-cell RNA-seq (*Schwarz et al., 2012*). Nine individual AWC neurons were dissected, separately amplified for their transcripts with RT-PCR, and equal aliquots from the nine separate purified products were sequenced as a single AWC RT-PCR pool. In order to obtain independent biological replicates for statistical analysis, we then also sequenced five aliquots of five individual RT-PCR products from five individually dissected AWC neurons.

We detected expression of 5894 protein-coding genes and 43 non-protein-coding genes that had a minimum estimated expression level of at least 0.1 transcripts per million (TPM) with a credibility interval of 99% (minTPM). The overall pattern of gene activity in AWC neurons is shown in *Figure 7*; details of observed gene activity, with summaries of gene function are given in *Figure 7—source data 1*. This AWC single-cell RNA-seq data set is not perfect; for example, some of the known AWC$^{on}$ markers such as *str-2* were not detected in our analyses while other genes were not known to express in the AWC$^{on}$ were found in our RNA-seq data set, which is likely due to that the RNAseq analysis having a higher sensitivity than the GFP reporter assays.

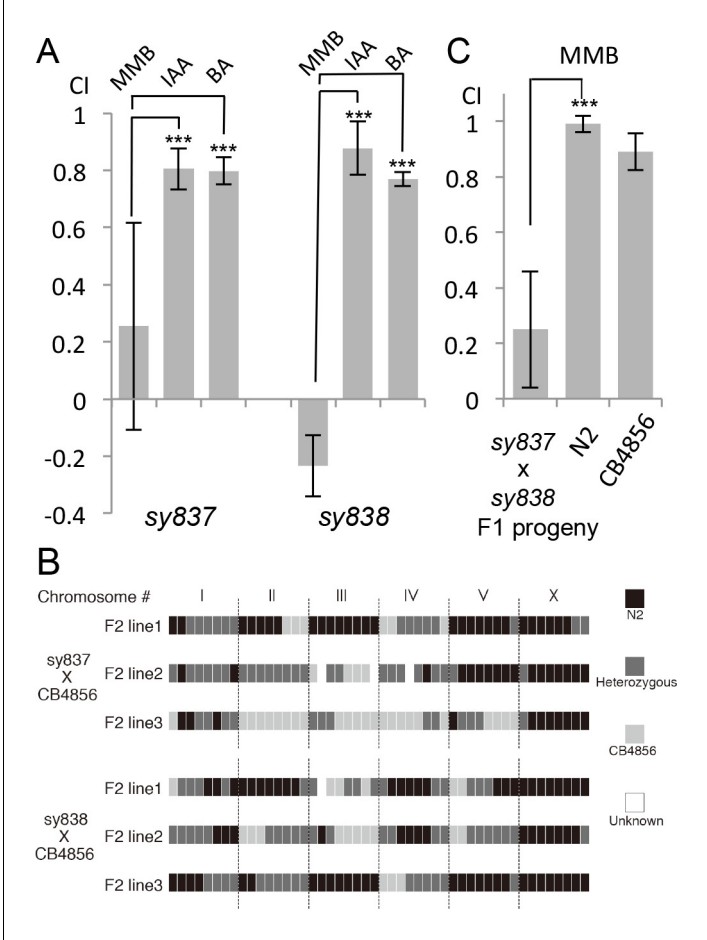

**Figure 6.** Forward genetic screens isolated two mutants with strong defects in MMB chemotaxis. (**A**) Response of the two mutant lines to MMB, IAA, and Benzaldehyde (BA; n = 4–8 trials for each odorant tested). The mutants showed strong defects in MMB chemotaxis but remained attracted to two other odorants, IAA and BA. (**B**) SNP mapping with the Hawaiian polymorphic strain CB4856 revealed that the mutations in both mutants were located on chromosome X. Mutants #7–22 and #18–10 were crossed to CB4856 and three F2 progeny from each mutant line that had MMB chemotaxis defects were analyzed for eight SNP markers on each chromosome. (**C**) Mutants #7–22 and #18–10 failed to complement. Responses of the F1 progeny from a cross between the two mutants to $10^{-4}$ dilution of MMB. (Mean ± SD, n = 5–8 trials).

The following figure supplements are available for figure 6:

**Figure supplement 1.** Gene structure of *odr-7*.

**Figure supplement 2.** Fosmid clone WRM0639bF05 which contains *odr-7* rescues the MMB attraction defect of *odr-7* (*sy837*).

Gene Ontology analysis showed that genes specifically upregulated in AWC neurons preferentially encoded a number of functions, including G-protein-coupled receptor activity [GO:0004930] and olfactory learning [GO:0008355] (*Figure 7—source data 3*). Among the 5894 genes, we identified 48 GPCRs, of which 34 encoded one of the putative chemosensory receptor PFAM domains (*Figure 7—source data 4*). In an earlier analysis of a subset of our RNA-seq data by older computational methods, we identified 46 GPCRs, of which 35 encoded a chemosensory PFAM domain (*Figure 7—source data 4* and *Figure 7—source data 2*). We speculated that some of these receptors might function as the chemoreceptors for the odors produced by *A. oligospora* that evoke a strong response in the AWC neurons.

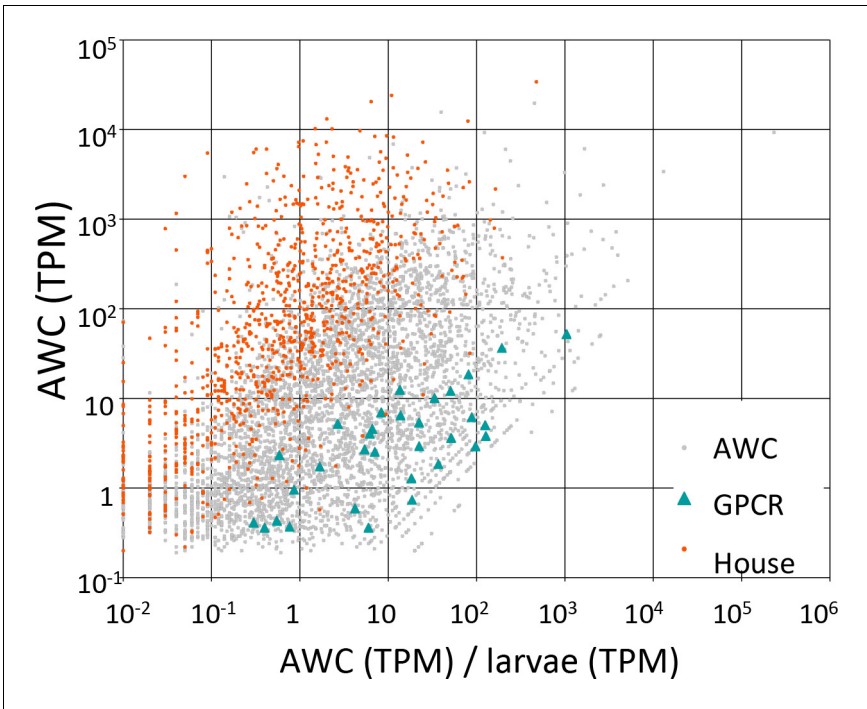

**Figure 7.** Candidate GPCRs expressed in AWC neurons. Gene expression in AWC neurons. Expression data are shown for 5326 *C. elegans* genes (5294 protein-coding and 32 non-coding RNAs) that exhibited above-background expression in our pooled AWC RNA-seq (i.e., that had a minimum expression level, in a 99% credibility interval, of ≥0.1 TPM). The x-axis shows AWC-specificity, computed as the ratio of AWC gene expression (in TPM) to larval gene expression (in TPM) for each gene. The y-axis shows the absolute magnitude of AWC gene activity in TPM. Among most genes (labeled 'AWC' and shown as gray dots) are highlighted 32 genes encoding G-protein coupled receptors shown as green triangles and 984 genes encoding housekeeping functions shown as red dots. The housekeeping genes were identified in a previous single-cell RNA-seq analysis (*Schwarz et al., 2012*). Full identities, annotations, and expression data for all *C. elegans* genes are provided in *Figure 7—source data 1*.

The following source data is available for figure 7:

**Source data 1.** Traits of *C. elegans* genes expressed in AWC, with RNA-seq expression data generated with our full data set and our most recent computational methods.

**Source data 2.** Traits of *C. elegans* genes expressed in AWC, with RNA-seq expression data generated with our older data set and computational methods.

**Source data 3.** Gene functions associated with genes specifically expressed in AWC.

**Source data 4.** Comparison of selected data for *C. elegans* genes encoding GPCRs expressed in AWC.

## Discussion

For a non-motile predator like *A. oligospora*, catching fast-moving nematode prey is a major task. The fact that in *A. oligospora* trap morphogenesis is usually only triggered by the presence of nematodes makes it even more challenging. Since trap morphogenesis takes more than 12 hr, there is a long period during which nematodes could escape before the functional traps become fully developed. In this study, we have shown that *A. oligospora* may overcome this problem by luring nematodes to stay nearby, by producing several volatile compounds that mimic nematode food and sex cues.

In nature, *Caenorhabditis* nematodes are most commonly associated with rotten fruits and plants where they feast on the bacterial blooms in these decomposing materials (*Frézal and Félix, 2015*).

Several of the odors we identified from *A. oligospora*, including the sulfurous compounds DMDS, DTP and SMT, have pungently overripe or rotten smells, suggesting that these odors might be sensed as food cues by nematodes and, as a result, are attractive to *C. elegans*. Such olfactory mimicry is also employed by various plants to attract pollinators. For example, the dead horse arum (*Helicodiceros muscivorus*) and the infamous corpse flower (*Amorphophallus titanium*) are known to produce rotten smells to attract insects; in both cases, one of the odors identified was DMDS (*Lim, 2012*; *Shirasu et al., 2010*; *Stensmyr et al., 2002*). Similarly, carnivorous plants such as the Venus flytraps, sundews and pitcher plants produce volatiles that mimic fruits and flowers to attract insect prey (*Di Giusto et al., 2010*; *Jurgens et al., 2009*). These examples demonstrate that olfactory mimicry of food might be a common strategy used by carnivorous plants and fungal species to lure their prey.

MMB, the most attractive compound we identified from *A. oligospora* volatiles, elicited a strong sex- and developmental stage-specific attraction for many *Caenorhabditis* nematodes. In the gonochoristic *Caenorhabditis* species (*C. remanei*, *C. nigoni* and *C. afra*), MMB was highly attractive to the females but repulsive to the males. In the hermaphroditic *Caenorhabditis* species including *C. elegans* and *C. tropicalis*, MMB was highly attractive to the adult hermaphrodites, but not to the males, dauers, or larvae. The presence of MMB strongly interfered with mating in *C. afra*, which is in contrast to the presence of diacetyl, another attracting chemical that is likely to be a food cue. If the female nematodes were adapted to MMB, mating was also strongly affected. Therefore, we hypothesize that MMB might be an olfactory mimic of a sex pheromone produced by male nematodes.

Multiple species have been reported to produce pheromone mimics to attract prey or pollinators. For example, the American bolas spider *Mastophora hutchinsoni* produces odorants that mimic female moth sex pheromones to attract different moth species (*Haynes et al., 2002*). Orchids such as *Ophrys sphegodes*, *O. exaltata* and *Mormolyca ringens* produce volatile compounds that greatly resemble the chemical composition of female bee pheromones in order to attract the male bees as pollinators (*Flach et al., 2006*; *Mant et al., 2005*; *Schiestl et al., 2000*). Whether male *Caenorhabditis* nematodes produce any volatile sex pheromones is currently unknown, but a recent study has found that male pinewood nematodes *Bursaphelenchus xylophilus* produce an as yet unidentified volatile pheromone that attracts the females (*Shinya et al., 2015*). Our study suggests that MMB produced by *A. oligospora* might function as an olfactory mimic of a sex pheromone produced by male nematodes to attract hermaphrodites or females. Therefore, it would be of interest to screen male *Caenorhabditis* nematodes for the production of any volatile compounds that could be functionally or structurally related to MMB. Why *A. oligospora* produces an odor that only attracts the females and the hermaphrodites is open to speculation. We suspect that in the niches inhabited by *A. oligospora*, there is a larger population of female and hermaphroditic nematodes than males. Furthermore, males are potently attracted to females and hermaphrodites so they might be also lured to the trapped female and hermaphroditic nematodes. *C. elegans* dauers are not attracted to MMB or *A. oligospora*, which might enable them to better survive *A. oligospora* predation. MMB attraction only seems to be conserved in *Caenorhabditis* species; *P. pacificus* was attracted to *A. oligospora*, but not to MMB, suggesting that *A. oligospora* attraction is likely mediated via other odorants in *P. pacificus*.

*A. oligospora* has evolved a strategy of olfactory mimicry to efficiently attract prey, but is this strategy common among other species of predacious fungi? Most known nematode-trapping fungi, like *A. oligospora*, only generate traps upon sensing the presence of nematodes, so they may benefit by attracting their prey. We believe that the olfactory mimicry strategy used by *A. oligospora* likely represents a conserved mechanism that is employed by many other predacious fungi. In fact, an earlier study reporting that the nematode *Panagrellus redivivus* was attracted to another *Arthrobotrys* species supports this hypothesis (*Balan and Gerber, 1972*).

In *C. elegans*, the olfactory neuron AWCs are known to mediate chemoattraction to certain volatile compounds (*Bargmann et al., 1993*; *Bargmann and Horvitz, 1991*). AWC neurons have also recently been shown to function in thermosensation and to respond to several other types of sensory stimuli, indicating that this neuron plays multiple roles in sensing environments and modulating *C. elegans* behavior (*Kuhara et al., 2008*; *Zaslaver et al., 2015*). We believe that our single-cell RNA-seq analysis of AWC neurons provides a useful transcriptomic dataset with which the *C. elegans* community can study diverse functions of AWC. Our study has demonstrated that AWC neurons can sense different odors produced by *A. oligospora* (*Figure 4A*), and the response is likely to be primary, as strong activation of the AWC$^{on}$ neuron was still observed in the *unc-13* mutant (*Figure 4B*). We have identified at

least 34 chemosensory GPCRs expressed in the AWC neurons, many of which might be candidates for receptors of the *A. oligospora* odors. *A. oligospora* attracts a broad range of nematode species, and we suspect that this is due to the production of a multitude of attractive odorants that are attractive to different nematode species and are sensed by multiple chemoreceptors in the nematodes. As a consequence, attraction to *A. oligospora* has been maintained in nematodes throughout evolution despite the strong selective pressure against it.

In conclusion, our data has shown that *A. oligospora* has evolved to produce olfactory mimics of food and sex cues to lure its nematode prey, revealing a novel molecular mechanism for microscopic predator-prey interactions (*Figure 8*). We expect that this example of coevolution will provide a foundation for future work that will further elucidate how different nematodes interact with different predatory fungi, while having broader implications for understanding other types of predator-prey interactions and their coevolution.

## Materials and methods

### Strains

All nematodes used in this study were maintained under standard conditions at 20°C. *C. elegans* strains used in this study included N2, CB4856 (Hawaiian), CX2065 *odr-1(n1936)*, CX2205 *odr-3 (n2150)*, CB3329 *che-10(e1809)*, PR678 *tax-4(p678)*, CX3085 *tax-2(ks31); tax-4(p678)*, CX4 *odr-7(ky4)*, PR813 *osm-5(p813)*, NL333 *gpa-1(pk15)*, PS6663 *syEx1324[Pstr-2::GCaMP6s::unc-54 3'UTR; Pofm-1::*

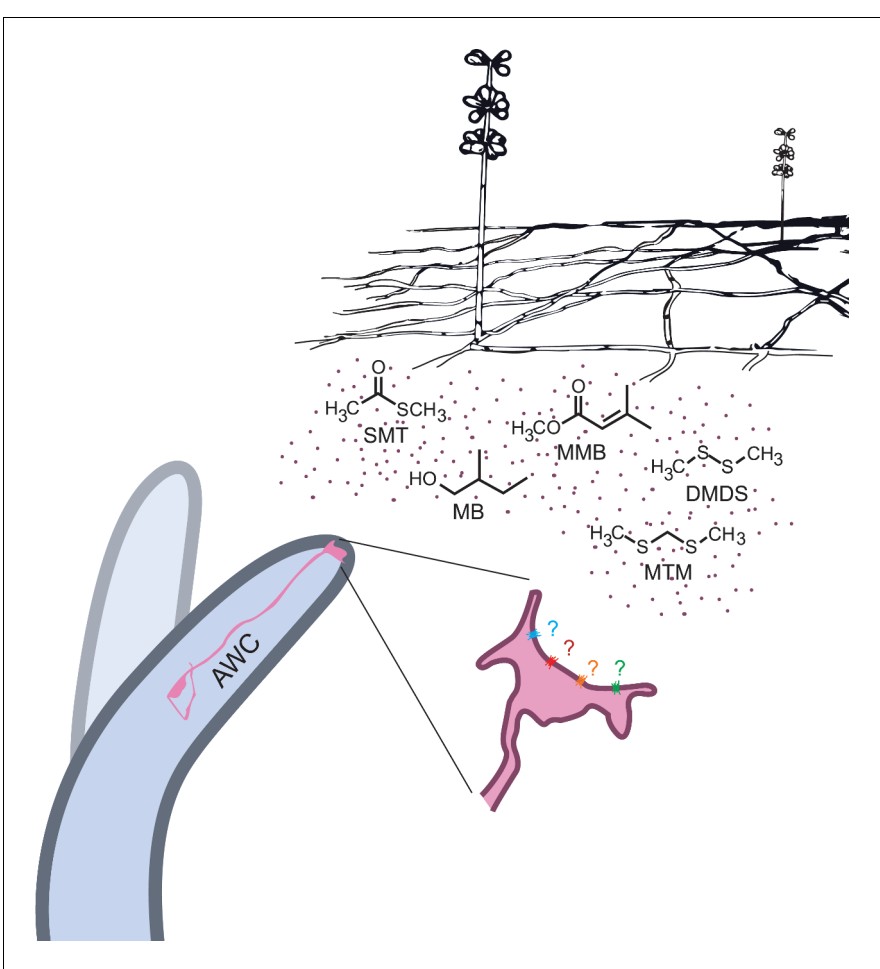

**Figure 8.** *A. oligospora* produce volatiles that mimic food and sex cues to attract *C. elegans* via its olfactory neuron AWCs.

*RFP]*, PS6712 *syEx1336[Pstr-2::GCaMP6s::unc-54 3'UTR; Pofm-1::RFP] unc-13(e51)*; CX3695 *kyIs140 [str-2::GFP + lin-15(+)]*, PS7117 *odr-7(sy837)*, PS7117 *odr-7(sy837)*, and TWN1 *odr-7(sy837)*; *yphEx1 [WRM0639bF05+ ofm-1::RFP]*. Different nematode species used in the chemotaxis assays were: PS1010 *C. angaria* (*sp. 3*), EG4788 *C. portoensis* (*sp. 6*), JU1199 *C. afra* (*sp. 7*), JU1325 *C. nigoni* (*sp. 9*), JU1373 *C. tropicalis* (*sp. 11*), JU1426 *C. castelli* (sp. 12), AF16 *C. briggsae*, PB4641 *C. remanei*, PS312 *Pristionchus pacificus,* and MT8872 *Panagrellus redivivus*. The fungal strains used in this study were *Arthrobotrys oligospora* (CBS115.81), *Aspergillus terreus* (NIH2624) and *Aspergillus niger* (ATCC11414).

## Chemotaxis assays

Unless noted otherwise, chemotaxis assays were population assays performed with synchronized adult animals in the presence of sodium azide as described previously (*Bargmann et al., 1993*). The responses of different nematode species to *A. oligospora* were evaluated with a 4-point assay as described in *Figure 1A*. Chemotactic responses to different chemicals were tested using the standard 2-point assays (*Bargmann et al., 1993*). Chemicals were diluted $10^{-2}$ for chemotaxis assays unless specified otherwise. The results were subjected to standard t-tests for statistical significance.

## Headspace sampling

Headspace volatiles were sampled using solid-phase microextraction (SPME). Prior to collection, each SPME fiber (divinylbenzene/carboxen/polydimethylsiloxane, 50/30 mm; Supelco, Bellefonte, PA) was pre-conditioned for 10 min at 275°C in the GC injector. The SPME fibers were inserted through the vial septa and exposed for a time period of 12–48 hr at room temperature, and then analyzed immediately.

## GC X GC TOF MS analyses

Comprehensive two-dimensional gas chromatographic mass spectral analyses (GC x GC-TOFMS) were carried out using a Pegasus 4D (Leco Corp., ST. Joseph, MI) equipped with an Rxi-5ms primary column (5% diphenyl/95% dimethyl polysiloxane; 30m length x 250 mm i.d. x 0.25 mm film thickness; Restek, Bellefonte, PA) and an Rxi-17 secondary column (50% diphenyl/50% dimethyl polysiloxane; 0.8 m length x 180 mm i.d. x 0.18 mm film thickness; Restek, Bellefonte, PA). The primary column oven temperature was held at 40°C for 1 min, then increased at a rate of 3 °C/min to a temperature of 150°C, after which it was increased at a rate of 12 °C/min to a final temperature of 275°C. The secondary column oven followed a temperature program that maintained a + 15°C offset from the primary column oven. A 3.5 s modulation period (0.80 hot pulse; 0.95 cold pulse) was used with a modulator temperature program which initiated at 105°C for 1 min then increased at 5 °C/min to a final temperature of 275°C. Splitless injection was used, with the split valve opening at 20s. The carrier gas flow rate was 1.3 mL/min.

## Calcium imaging

Calcium imaging was performed in a microfluidic olfactory chip designed by the Bargmann lab (*Chronis et al., 2007*). Adult animals expressing GCaMP6 under the *str-2* promoter in the wild-type (N2) or the *unc-13*(e51) background were used for imaging. Plasmid pSM was used as the backbone plasmid for cloning $P_{str-2}$::GCaMP6 (*Mccarroll, 2005*). Animals were presented with alternating streams of S-basal buffer and different fungal odorants diluted in S-complete buffer. Image analysis was done using a custom script written in Python that calculated fluorescence intensity from an individual region of interest (ROI) (*Cho and Sternberg, 2014*).

## Mating behavior of *C. afra*

*C. afra* was maintained on OP50 and NG medium as for *C. elegans*. To track the mating behavior of the nematodes, L4 male and female animals were picked onto separate plates 24 hr before tracking. One female and two male adults were then placed on the opposite sides of a 3 cm chemotaxis plate and recorded immediately until mating occurred or 30 min if no mating behavior was observed.

## Genetic screen, SNP mapping and whole-genome sequencing

N2 animals were treated with ethyl methanesulfonate (EMS) following a standard protocol (*Kutscher and Shaham, 2014*). Semi-synchronized F2 populations were subjected to the chemotaxis screens. Mutants that chemotacted to IAA, instead of MMB, were collected and subjected to another round of assays. Approximately 40,000 genomes were screened, and about a dozen mutants were isolated. Two mutants that exhibited a strong phenotype were subjected to further analyses. SNP mapping was done using methods developed by the Jorgensen lab (*Davis et al., 2005*). Whole-genome sequencing data were analyzed by the CloudMap pipeline developed by the Hobert lab (*Minevich et al., 2012*).

## Single cell transcriptomic analyses of the AWC neuron

GFP-labeled AWC^on cells from the *C. elegans* strain CX3695 (provided by the Bargmann lab) were micro-dissected and were frozen immediately in liquid nitrogen. Cells were lysed gently and subjected to 3'-tailed RT-PCR based on a method developed to amplify cDNA from a single cell (*Schwarz et al., 2012*). We dissected nine individual AWC neurons, separately amplified their transcripts with RT-PCR, and purified and quantitated the RT-PCR products before sequencing them. We first removed equal aliquots from the nine separate purified products into a single AWC RT-PCR mixture, and sequenced the mixture as a single AWC RT-PCR pool. In order to obtain independent biological replicates for statistical analysis, we then also sequenced five aliquots of five individual RT-PCR products from five individually dissected AWC neurons.

As a control for detecting the activities of genes with housekeeping and other functions not specific to AWC, we reused two published RNA-seq read sets, generated by RT-PCR of two separate dilute aliquots from a single preparation of *C. elegans* whole animal RNA (*Schwarz et al., 2012*). The preparation was obtained from starved mixed-stage hermaphrodites, mostly L3 and L4 stage larvae, with some young adults and L2-stage larvae, and a few L1-stage larvae and eggs; as before, we briefly describe it as 'larval RNA'. This larval RNA had previously been quantitated and tested for quality with a Bioanalyzer, and successfully used for standard full-length RNA-seq (*Mortazavi et al., 2010*). The two RNA-seq read sets counted as technical rather than biological replicates, because their starting RNA aliquots derived from a single whole larval RNA stock. However, single-cell RT-PCR amplification is an inherently noisy procedure for both technical and biological reasons (*Davidson et al., 2014*), and the different expression values for genes in the two larval RNA-seq read sets therefore provided important information about the noisiness of comparisons between expression levels of genes in AWC neurons versus whole animals. We thus treated these two larval read sets as independent replicates in our statistical analysis (below), while pooling them into a single data set for estimates of gene activity in whole larvae.

We did two rounds of RNA-seq analysis. Our initial round of RNA-seq analysis was done by methods essentially identical to those in *Schwarz et al. (2012)*. In particular, we mapped raw RNA-seq reads to *C. elegans* protein-coding genes from the WS190 release of WormBase with bowtie (*Langmead et al., 2009*), and used ERANGE 3.1 (*Mortazavi et al., 2008*) to quantitate their mapped read counts per gene and their gene expression levels in Reads per Kilobase of exon model per Million mapped reads (RPKM). We carried out this initial analysis using only the RNA-seq data from our pool of nine AWC cells, along with our previously published whole-larval RNA-seq data as a control for housekeeping and non-specific genes. RPKM values from this analysis (mapped to current gene names in WormBase), along with current gene annotations, are given in *Figure 7—source data 2*.

Subsequently, as we added more data, both our sequencing technologies for the AWC RT-PCRs and our RNA-seq analysis methods became more advanced than they had been in *Schwarz et al. (2012)*. For RNA-seq, although the first round of AWC RNA-seq had raw 38-nt sequencing reads as before, the second round had raw 50-nt reads instead, and their FastQ quality coding scale (*Cock et al., 2010*) changed between the first and second AWC sequencing runs. To allow coherent use of all AWC reads as a single pooled data set for quantitating AWC gene expression, we recoded the quality scale of the older AWC read set (fastq-illumina, phred score with offset of 64) to that of the newer (fastq-sanger, phred score with offset of 33) using *seqret* from EMBOSS 6.5.7 (*Rice et al., 2000*) before further analysis.

After RNA-seq of the AWC cells, we likewise used more advanced methods to analyze the RNA-seq data. All reads were quality-filtered as follows: AWC reads that had failed Chastity filtering were discarded (Chastity filtering had not been available for the larval reads); raw 38-nt reads were trimmed 1 nt to 37 nt; all reads were trimmed to remove any indeterminate ('N') residues or residues with a quality score of less than 3; and raw 38-nt reads that had been trimmed below 37 nt were deleted, as were raw 50-nt AWC reads that had been trimmed below 50 nt. This left us with a total of 131,614,334 filtered AWC reads for analysis, of which 96,433,929 reads could be partitioned into biologically independent replicates (*Supplementary file 2A*).

Proceeding to RNA-seq analysis proper, we used RSEM (version 1.2.17; *Li and Dewey, 2011*, PMID 21816040) with bowtie2 (*Langmead and Salzberg, 2012*) and SAMTools (*Li et al., 2009*) to map filtered reads to a *C. elegans* gene index and generate readcounts and gene expression levels in transcripts per million (TPM). To create the *C. elegans* gene index, we ran RSEM's *rsem-prepare-reference* with the arguments '*–bowtie2 –transcript-to-gene-map*' upon a collection of coding DNA sequences (CDSes) from both protein-coding and non-protein-coding *C. elegans* genes in Worm-Base release WS245 (*Harris et al., 2014*). The CDS sequences were obtained from *ftp://ftp. sanger. ac. uk/ pub2/ wormbase/ releases/ WS245/ species/ c_ elegans/ PRJNA13758/ c_ elegans. PRJNA13758. WS245. mRNA_ transcripts. fa. gz* and *ftp:// ftp. sanger. ac. uk/ pub2/ wormbase/ releases/ WS245/ species/ c_ elegans/ PRJNA13758/ c_ elegans. PRJNA13758. WS245. ncRNA_ transcripts. fa. gz*.

To analyse overall gene expression levels in AWC neurons, we pooled all 132 million RNA-seq reads from both the nine-cell pool and the five individual cells into a single AWC read set before mapping them and computing their genewise TPMs. Similarly, to analyse overall gene expression levels in whole larvae for comparison to AWC neurons, we pooled all 23.4 million RNA-seq reads from both larval data sets into a single larval data set before mapping it and computing its TPMs.

Conversely, to analyze the statistical significance for differences of gene expression between AWC neurons and whole larvae, we used only the read sets from the individual AWC neurons. Our past experience with single-cell RNA-seq data sets (*Schwarz et al., 2012*) led us to expect that gene expression data from single AWCs would be highly noisy, but that this noise would be canceled out by pooling non-overlapping subsets of the individual AWCs and analyzing them as independent biological replicates. We thus mapped and quantitated the single-cell AWC read sets as two pools (cells 1, 2, and 4 for pool 1; cells 3 and 5 for pool 2). Cells were chosen for the two pools so that the total number of reads in the two pools would be as close to equal as possible (45.7 million and 50.7 million reads for pools 1 and 2; *Supplementary file 2A*).

For each RNA-seq data set of interest, we computed mapped reads and expression levels per gene by running RSEM's *rsem-calculate-expression* with the arguments "*–bowtie2 -p 8 –no-bam-output –calc-pme –calc-ci –ci-credibility-level 0.99 –fragment-length-mean 200 fragment-length-sd 20 –estimate-rspd –ci-memory 30000*". These arguments, in particular '*–estimate-rspd*', were aimed at dealing with single-end data from 3'-biased RT-PCR reactions; the arguments '*–phred33-quals*' and '*–phred64-quals*' were also used for the AWC and larval reads, respectively. We computed posterior mean estimates both for read counts and for gene expression levels, and rounded PMEs of read counts down to the nearest lesser integer. We also computed 99% credibility intervals (CIs) for expression data, so that we could use the minimum value in the 99% CI for TPM as a reliable minimum estimate of a gene's expression (minTPM). We defined non-zero, above-background expression for a gene in a given RNA-seq data set by that gene having an minimum estimated expression level of at least 0.1 transcripts per million (TPM) in a credibility interval of 99% (minTPM). The most relevant results from RSEM analysis are given in *Figure 7—source data 1*. The numbers of genes being scored as expressed in AWC above background levels, for various data sets, are given in *Supplementary file 2B*.

For individual genes, we computed the significance for differences of gene activity between AWC and whole larvae with DESeq2 (*Love et al., 2014*). DESeq2 was run in R 3.2.3, using default parameters to compute p-values that had been adjusted for multiple testing (*Noble, 2009*) by the collective false discovery rate (FDR) formula (*Benjamini and Hochberg, 1995*). We computed these FDR-adjusted p-values ('padj') by testing two pools of individual AWC read sets against two larval sets, at an $\alpha$ threshold of 0.01; genes for which this gave significant changes are listed in *Figure 7—source data 1*. We defined genes expressed significantly more strongly in AWC than in whole larvae as those genes with padj $\leq$ 0.01 and AWC/larvae expression ratios of >1.

## Data availability

RNA-seq reads for the wild-type AWC neurons are available in the NCBI Sequence Read Archive (SRA), under accession number SRP074082 (http://trace.ncbi.nlm.nih.gov/Traces/sra/?study=SRP074082). RNA-seq reads for the two pools of whole *C. elegans* mixed-stage wild-type N2 larvae were previously published by *Schwarz et al. (2012)*, and are available in the NCBI SRA under accession number SRP015688 (http://trace.ncbi.nlm.nih.gov/Traces/sra/?study=SRP015688).

## Acknowledgements

We thank Marie-Anne Felix (IBENS), Cori Bargmann (The Rockefeller University), Ryoji Shinya (Sternberg lab), Vivian Chiu (then in Sternberg lab) and the Caenorhabditis Genetics Center (University of Minnesota) for providing the nematode strains used in this study. We also thank Valerie Knowlton (NCSU) for acquiring the SEM image and Clay Wang (USC) for providing the *Aspergillus* strains. Michael Milligan (SUNY Fredonia) provided invaluable assistance with GC x GC-TOFMS analyses and Chris Cronin and Sreekanth Chalasani (Salk Institute) provided advices for the calcium imaging set-up. We thank Gladys Medina, Sarah Kim, Chiang Yu Shi and Ka-Naam Heung for providing technical assistance and members of the Sternberg laboratory for helpful discussion. This work was supported by HHMI, with which PWS is an investigator, by the NIH K99 award 1 K99GM108867-1 to YPH, by NIH GM084389 to PWS, and startup funds from Academia Sinica to YPH. RN was supported by T32-GM007616.

## Additional information

### Funding

| Funder | Grant reference number | Author |
| --- | --- | --- |
| National Institute of General Medical Sciences | 1K99GM108867-1 | Yen-Ping Hsueh |
| Academia Sinica | | Yen-Ping Hsueh |
| National Institutes of Health | T32-GM007616 | Paul W Sternberg |
| Howard Hughes Medical Institute | 047-101 | Paul W Sternberg |
| National Institutes of Health | GM084389 | Paul W Sternberg |

The funders had no role in study design, data collection and interpretation, or the decision to submit the work for publication.

### Author contributions

Y-PH, Conceptualization, Data curation, Formal analysis, Supervision, Funding acquisition, Validation, Investigation, Writing—original draft, Writing—review and editing; MRG, Data curation, Formal analysis, Writing—original draft; EMS, Data curation, Formal analysis, Methodology, Writing—original draft, Writing—review and editing; RDN, C-HL, Data curation, Formal analysis; SG, Data curation, Validation; FCS, Formal analysis, Supervision, Writing—review and editing; PWS, Conceptualization, Resources, Supervision, Funding acquisition, Writing—review and editing

### Author ORCIDs

Paul W Sternberg, http://orcid.org/0000-0002-7699-0173

## Additional files

### Supplementary files

• Supplementary file 1. Detection of compounds 1-5 from headspace above fungal cultures of *A. oligospora*, *A. terreus*, *A. niger*, and growth medium-only control samples.

• Supplementary file 2. Summary of RNA-seq data and expressed gene counts.

**Major datasets**

The following dataset was generated:

| Author(s) | Year | Dataset title | Dataset URL | Database, license, and accessibility information |
|---|---|---|---|---|
| Hsueh YP, Gronquist MR, Schwarz EM, Nath RD, Lee CH, Gharib S, Schroeder FC, Sternberg PW | 2016 | Single-cell RNA-seq of AWC[on] chemosensory neurons of Caenorhabditis elegans | https://trace.ncbi.nlm.nih.gov/Traces/sra/?study=SRP074082 | Publicly available at the NCBI Sequence Read Archive (accession no. SRP074082) |

The following previously published datasets were used:

| Author(s) | Year | Dataset title | Dataset URL | Database, license, and accessibility information |
|---|---|---|---|---|
| Schwarz EM, Kato M, Sternberg PW | 2012 | C. elegans linker cell transcriptome | https://trace.ncbi.nlm.nih.gov/Traces/sra/?study=SRP015688 | Publicly available at NCBI Sequence Read Archive (accession no. SRP015688) |

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
