## [Decision Letter]

Thank you for submitting your article "Fatal attraction: nematophagous fungus *A. oligospora* mimics olfactory cues of sex and food to lure nematodes" for consideration by *eLife*. Your article has been reviewed by two peer reviewers, and the evaluation has been overseen by a Reviewing Editor and a Senior Editor. The following individuals involved in review of your submission have agreed to reveal their identity: Leslie B Vosshall (Reviewer #1); Douglas Portman (Reviewer #2).

The reviewers have discussed the reviews with one another and the Reviewing Editor has drafted this decision to help you prepare a revised submission.

This manuscript demonstrates that several species of nematodes are attracted to a nematode-trapping fungus, *A. oligospora*. A single volatile produced by *A. oligospora*, MMB, is sufficient to attract females or hermaphrodites from many nematode species. In hermaphrodite *C. elegans*, this attraction requires AWC sensory neurons, and AWC neurons respond to many odors produced by *A. oligospora*, including MMB. The description and identification of volatiles that attract *C. elegans* is interesting, and provides a platform for further study of micro predator-prey relationships.

Major comments:

1) Of greatest concern is that many of the data are apparently not subjected to any statistical analysis. If results are non-significant in all panels without reported statistics, the majority of the paper's claims are unsupported. It is troubling that the only statistics in the paper have ~6 times higher Ns than other experiments. This should be easily addressed.

2) The sex pheromone mimicry was criticized by both reviewers, phrased in the following manner:a) The claim in the text and the title that MMB mimics a sex pheromone only based on its selective effect on adult female hermaphrodites is very speculative. I do not have an issue with equating MMB with a food cue because the GC-MS-identified volatiles are plausible microbial metabolites, but nothing that the fungi produce looks like a worm pheromone. Further, AWC has never been described as a pheromone-sensing neuron.

b) The idea that MMB mimics sex pheromones is a major point of the paper. While the rationale for the production of food-like cues is good, I'm not entirely convinced that the authors have demonstrated that MMB mimics a sex pheromone. The authors' conclusion stems from two observations: first, that hermaphrodites are more strongly attracted to MMB than are males, and second, that hermaphrodite attraction is adult-specific. With regard to the first point, it's been recently shown that hermaphrodites are significantly more attracted than males to the food cue diacetyl and to bacterial food itself (Ryan et al. Curr Biol 2014). Thus, sex specificity alone is not conclusive evidence (i.e., we wouldn't call diacetyl a sex pheromone). If MMB is truly mimicking a male sex pheromone, there are several predictions – for example, the presence of MMB might interfere with hermaphrodite responses to genuine male pheromones, and srx-96 mutant hermaphrodites should be less attracted to males. The problem with this, as the authors note, is that there's little evidence in *C. elegans* that hermaphrodites are attracted to male pheromones at all – the only relevant evidence I know of is from the Schroeder lab (Izrayelit et al. 2012 ACS Chem Biol). Certainly, it could be that the more relevant function of MMB is to mimic male sex pheromones of related (likely male/female) nematode species, but evaluating this would not be straightforward. With regard to the second point (adult-specificity), Figure 1 pretty clearly indicates that L1 animals are indeed attracted to *A. oligospora*. It's true that dauers are not attracted, but given how little we know about dauer sensory behavior, I'm not sure this is particularly informative.

Both reviewers recommend to rewrite the paper to remove this speculation – this requires a change to every part of the paper, including the title and removing the extensive pheromone speculation in the Discussion. It is an interesting observation, but the elucidation mechanism would require a lot more work.

3) The text identifies mutations in *odr-7* as the likely locus responsible for the decrease in chemotaxis, but when compared to *odr-7* mutants in Figure 2, the EMS mutants appear to have a stronger phenotype. To claim that *odr-7* is the source of MMB loss, a rescue experiment is required, or at least failure to complement with *odr-7* mutants. It is similarly necessary to show Hawaiian chemotaxis to MMB to make the mapping results interpretable. Further I don't understand the logic that *odr-7* mutants are necessarily affecting only chemosensory receptor expression as the underlying mechanism for perturbations in MMB sensing. Couldn't it be any number of other transcriptional targets of *odr-7* that would affect the behavior (e.g. downstream signaling components specific to AWC)?

4) The *srx-96* phenotype is very weak and no rescue experiments are offered to show that this defect maps to AWC. It is obvious that the very large number of chemosensory receptors in the worm make it very hard to get big phenotypes, but pulling back on the interpretation is warranted unless the authors can do more with this receptor.

5) A central conclusion of the paper is: "*A. oligospora* has evolved to lure the nematodes by producing olfactory mimicry of their food and sex cues which are attractive to Caenorhabditis nematodes." Clearly *A. oligospora* produces multiple compounds that are attractive to worms. However, to conclude that it has evolved to lure nematodes by producing these compounds would seem to require some very specific evidence: namely, that these cues are produced specifically in order to attract prey, rather than being generated as metabolic (by-)products common to many fungi. I'm not convinced that the production of food-like cues is a specific adaptation that promotes predation by *A. oligospora*. These cues could more like generic secretions of many fungi to which *C. elegans* is attracted simply because they provide reliable signals of food sources. (The latter idea seems to be consistent with the production of MMB and other compounds by truffles.) In order to show that this is a "striking example of coevolution" (Discussion), I think the authors need more convincing evidence that the production of these signals is a trait that has arisen as a result of the predatory lifestyle. A step toward this might be to examine the relationship between predation and cue production in a family of related predatory and non-predatory fungi. Of course, this gets quite complicated: it's easy to imagine a scenario where a non-predatory ancestor produces some attractive cues. Since there are then a lot of nematodes in the neighborhood, predatory abilities are more likely to evolve. But would this mean that these cues are produced specifically to attract nematodes? Is this "striking coevolution"? I'm certainly no evolutionary biologist, so I'm open to persuasion here.

---

## [Author Response]

*Major comments:*

*1) Of greatest concern is that many of the data are apparently not subjected to any statistical analysis. If results are non-significant in all panels without reported statistics, the majority of the paper's claims are unsupported. It is troubling that the only statistics in the paper have ~6 times higher Ns than other experiments. This should be easily addressed.*

We have now included the results of the statistical analysis in the figures, and described how statistical analysis was conducted in the Materials and methods.

*2) The sex pheromone mimicry was criticized by both reviewers, phrased in the following manner:a) The claim in the text and the title that MMB mimics a sex pheromone only based on its selective effect on adult female hermaphrodites is very speculative. I do not have an issue with equating MMB with a food cue because the GC-MS-identified volatiles are plausible microbial metabolites, but nothing that the fungi produce looks like a worm pheromone. Further, AWC has never been described as a pheromone-sensing neuron.*

*b) The idea that MMB mimics sex pheromones is a major point of the paper. While the rationale for the production of food-like cues is good, I'm not entirely convinced that the authors have demonstrated that MMB mimics a sex pheromone. The authors' conclusion stems from two observations: first, that hermaphrodites are more strongly attracted to MMB than are males, and second, that hermaphrodite attraction is adult-specific. With regard to the first point, it's been recently shown that hermaphrodites are significantly more attracted than males to the food cue diacetyl and to bacterial food itself (Ryan et al. Curr Biol 2014). Thus, sex specificity alone is not conclusive evidence (i.e., we wouldn't call diacetyl a sex pheromone). If MMB is truly mimicking a male sex pheromone, there are several predictions – for example, the presence of MMB might interfere with hermaphrodite responses to genuine male pheromones, and srx-96 mutant hermaphrodites should be less attracted to males. The problem with this, as the authors note, is that there's little evidence in C. elegans that hermaphrodites are attracted to male pheromones at all – the only relevant evidence I know of is from the Schroeder lab (Izrayelit et al. 2012 ACS Chem Biol). Certainly, it could be that the more relevant function of MMB is to mimic male sex pheromones of related (likely male/female) nematode species, but evaluating this would not be straightforward. With regard to the second point (adult-specificity), Figure 1 pretty clearly indicates that L1 animals are indeed attracted to A. oligospora. It's true that dauers are not attracted, but given how little we know about dauer sensory behavior, I'm not sure this is particularly informative.*

*Both reviewers recommend to rewrite the paper to remove this speculation – this requires a change to every part of the paper, including the title and removing the extensive pheromone speculation in the Discussion. It is an interesting observation, but the elucidation mechanism would require a lot more work.*

We agree with the reviewer’s concerns about whether MMB truly mimics the sex pheromone and thus decided to conduct additional experiments to see if we can find more clues about how MMB affects the nematode behavior. As the reviewers suggested, if MMB is truly mimicking a male sex pheromone, the presence of MMB might interfere with the animal’s responses to genuine male pheromones. We therefore monitored the mating behavior of a male/female *Caenorhabditis* species (*C. afra*, also known as sp. 7) in the presence or absence of MMB and the results were included in the revision (Figure 5). We selected *C. afra* because it has highly robust and reproducible mating behavior. Using a worm tracker, we found that *C. afra* wild-type male and female animals would mate within an average of ~5 minutes after they were put on the assay plate. However, in the presence of MMB, the majority of the animals failed to mate during the entire tracking period (30 minutes). If the female worms were pre-exposed to MMB before mating, the majority of the animals also failed to mate (Figure 5). As a control, we also tracked mating behavior in the presence of another attractant that is more likely a food cue, diacetyl, and found that mating was not affected (Figure 5). We therefore think that this additional evidence further supports the idea that MMB might mimic a sex pheromone. We have also rephrased the wording in our manuscript, in order to make our interpretation of the data more nuanced and less speculative.

*3) The text identifies mutations in odr-7 as the likely locus responsible for the decrease in chemotaxis, but when compared to odr-7 mutants in Figure 2, the EMS mutants appear to have a stronger phenotype. To claim that odr-7 is the source of MMB loss, a rescue experiment is required, or at least failure to complement with odr-7 mutants. It is similarly necessary to show Hawaiian chemotaxis to MMB to make the mapping results interpretable. Further I don't understand the logic that odr-7 mutants are necessarily affecting only chemosensory receptor expression as the underlying mechanism for perturbations in MMB sensing. Couldn't it be any number of other transcriptional targets of odr-7 that would affect the behavior (e.g. downstream signaling components specific to AWC)?*

We generated a fosmid rescue line and tested the chemotaxis behavior. The defect can be rescued by the fosmid that contains the wild-type *odr-7* gene (Figure 6—figure supplement 2). The Hawaiian chemotaxis to MMB is now included (Figure 6). The screen was designed to avoid identifying the mutants in downstream signaling components, as these mutants were still attracted to another attractive compound, IAA. We did not think that the *odr-7* is affecting only the chemosensory receptor expression. Our logic was that since the only two mutants with strong defects came out from the screen were both *odr-7*, it is more likely that multiple receptors were involved in sensing this compound.

*4) The srx-96 phenotype is very weak and no rescue experiments are offered to show that this defect maps to AWC. It is obvious that the very large number of chemosensory receptors in the worm make it very hard to get big phenotypes, but pulling back on the interpretation is warranted unless the authors can do more with this receptor.*

We generated a genomic rescue line for the *srx-96* mutant, but it only partially rescued the phenotype. Therefore, we have removed the *srx-96* result and removed the gene name from our model figure.

*5) A central conclusion of the paper is: "A. oligospora has evolved to lure the nematodes by producing olfactory mimicry of their food and sex cues which are attractive to Caenorhabditis nematodes." Clearly A. oligospora produces multiple compounds that are attractive to worms. However, to conclude that it has evolved to lure nematodes by producing these compounds would seem to require some very specific evidence: namely, that these cues are produced specifically in order to attract prey, rather than being generated as metabolic (by-)products common to many fungi. I'm not convinced that the production of food-like cues is a specific adaptation that promotes predation by A. oligospora. These cues could more like generic secretions of many fungi to which C. elegans is attracted simply because they provide reliable signals of food sources. (The latter idea seems to be consistent with the production of MMB and other compounds by truffles.) In order to show that this is a "striking example of coevolution" (Discussion), I think the authors need more convincing evidence that the production of these signals is a trait that has arisen as a result of the predatory lifestyle. A step toward this might be to examine the relationship between predation and cue production in a family of related predatory and non-predatory fungi. Of course, this gets quite complicated: it's easy to imagine a scenario where a non-predatory ancestor produces some attractive cues. Since there are then a lot of nematodes in the neighborhood, predatory abilities are more likely to evolve. But would this mean that these cues are produced specifically to attract nematodes? Is this "striking coevolution"? I'm certainly no evolutionary biologist, so I'm open to persuasion here.*

In the subsection “Identification of *A. oligospora*-derived odorants that attract *C. elegans*”, we described the results of the volatile compounds produced by *A. oligospora*. Indeed, some of the compounds such as DMDS were commonly identified in different species including the truffles. We also detected DMDS in another two non-predatory ascomycetous species (*Aspergillus terreus* and *A. niger*). However, MMB is only present in *A. oligospora* but absent in the other two none-predatory fungi, demonstrating that MMB is not a general fungal odor that is produced by many fungal species. We think the production of MMB in *A. oligospora* is related to attracting the nematodes but agree that this may not be a “striking” example of coevolution. Thus, we have removed the word “striking” in the text, which is indeed unnecessary.